# Airsacculitis Caused by Enterobacteria and Occurrence of Eggs of the Superfamily Diplotriaenoidea in Feces of Tropical Screech Owl (*Megascops choliba*) in the Amazon Biome

**DOI:** 10.3390/ani13172750

**Published:** 2023-08-29

**Authors:** Hanna Gabriela da Silva Oliveira, Rafaelle Cunha dos Santos, Cinthia Távora de Albuquerque Lopes, Ananda Iara de Jesus Souza, Débora da Vera Cruz Almeida, Sarah Raphaella Rocha de Azevedo Scalercio, Aline de Marco Viott, Sheyla Farhayldes Souza Domingues, Felipe Masiero Salvarani

**Affiliations:** 1Instituto de Medicina Veterinária, Universidade Federal do Pará, Castanhal 68740-970, PA, Brazil; hnnagabriela@gmail.com (H.G.d.S.O.); rafaellesantosc@gmail.com (R.C.d.S.); cinthia@ufpa.br (C.T.d.A.L.); anandaiara@hotmail.com (A.I.d.J.S.); deboramedvet22@gmail.com (D.d.V.C.A.); shfarha@ufpa.br (S.F.S.D.); 2Centro Nacional de Primatas, Instituto Evandro Chagas, Ananindeua 67033-009, PA, Brazil; sarah.scalercio@fiocruz.br; 3Campus Palotina, Universidade Federal do Paraná, Palotina 85959-000, PR, Brazil; viott@ufpr.br

**Keywords:** *Escherichia coli*, *Klebsiella pneumoniae*, *Enterobacter aerogenes*, nematodes, air sacs, Strigiformes

## Abstract

**Simple Summary:**

Owls are widely distributed globally, and *Megascops choliba*, the tropical screech owl, is one of the most common species. There are reports of parasitic and bacterial infections in several species of raptors. However, information about infectious diseases in owls still needs to be made available, especially for this species, as most articles focus on its biology. This paper reports a case of airsacculitis caused by enterobacteria and the presence of eggs of the Diplotriaenoidea superfamily in *M. choliba*, which, as far as we know, has not been reported in the Brazilian and international literature so far. The clinical picture is described, including the therapy used, complementary exams, necropsy results, and histopathological findings.

**Abstract:**

This study aims to report the clinical signs, therapeutic strategy, necropsy results, and histopathological findings of airsacculitis caused by enterobacteria and the occurrence of eggs from the superfamily Diplotriaenoidea in the feces of *Megascops choliba* in the Amazon biome. A tropical screech owl nestling was rescued and admitted for hand-rearing. The animal was kept hospitalized for five months. It was fed a diet based on *Zophobas morio* larvae and thawed chicken breast meat with vitamin and mineral supplements. On the 37th day of hacking training for release, the owl showed weakness, lack of appetite, regurgitation, cachexia, dyspnea, ruffled feathers, dry droppings in the vent and pericloaca, and diarrhea. The parasitological examination showed eggs of the Diplotriaenoidea superfamily in the feces. The therapy employed included oxytetracycline, sulfamethoxazole, mebendazole, Potenay, sodium chloride 0.9%, and Mercepton. However, five days after starting the treatment, the bird died. Upon necropsy, prominence of the keel, pieces of undigested food in the oral cavity and proventriculus, intestinal gas, and thickened and turbid air sacs were found. The microbiological analysis of air sacs identified *Escherichia coli*, *Klebsiella pneumoniae*, and *Enterobacter aerogenes*. Histopathological examination showed heterophilic bacterial airsacculitis.

## 1. Introduction

Raptors are bird species that have retained their raptorial lifestyle derived from a common ancestor. They include species from the orders Accipitriformes, Cariamiformes, Cathartiformes, Falconiformes, and Strigiformes. Strigiformes is represented by owls [1], which are widely distributed throughout the world, with approximately 11% of all species occurring in Brazil, and most of them (23 species) are poorly studied [2]. Strigiformes currently comprise two distinct families: the Tytonidae and the Strigidae. Species of this order have variable sizes and adaptations for hunting in low-light environments and have well-developed, forward-facing eyes that enable binocular vision, in addition to very sensitive hearing for the location of prey and the outer area of the primary feathers adapted for silent flight. They generally have nocturnal and crepuscular habits, with some exceptions of dusk behavior [1,3].

The Strigidae family includes the genus Megascops, with *Megascops choliba*, popularly known as tropical screech owl, as one of the most frequently encountered birds of prey in rehabilitation centers in Brazil [4,5]. They are relatively small owls. Males measure between 20.6 cm and 30 cm and weigh from 80 g to 169 g, and females measure between 17.5 cm and 28 cm and weigh from 97 g to 196 g. The tropical screech owl’s diet includes arthropods such as grasshoppers, spiders, scorpions, and moths, along with mammals such as mice and bats; small reptiles and amphibians like frogs are also included [6].

Although this species is abundant in its distribution, information on bacterial and endoparasitic agents as causes of diseases affecting it within Brazil is scarce compared to that available for other Falconiformes and Strigiformes within the Amazon biome [4,5]. Among the predisposing factors for disease are nutritional deficiencies, environmental changes, concomitant diseases, stress in captivity, and inadequate sanitary management, all of which induce immunosuppression [7]. In addition, birds may develop different bacterial and parasitic infections without showing clinical signs, and when they do, they are often already in an advanced stage of the disease. This leads the animals to receive an unfavorable prognosis and often results in death [7].

Knowledge of the epidemiology of infectious agents and their relationships with potentially susceptible hosts is critical for assessing the risk of occurrence of a given pathology and its impact on biodiversity. In this context, determining the incidence and distribution of pathogens is of great urgency to know the actual sanitary status of captive and wild birds. And, carnivorous animals such as raptors, which occupy the top of the trophic network, can act as “bio accumulators” of pathogens, resulting in high infection rates and making them sentinels and strategic targets in surveillance programs for pathogen detection [5,7,8]. Therefore, given the importance of this information in birds of prey and the scarcity of data in the literature in this area, the objective of this study was to report the first case of airsacculitis caused by enterobacteria and eggs of the superfamily Diplotriaenoidea in *Megacops choliba* in the Amazon biome.

## 2. Materials and Methods

A tropical screech owl nestling (*Megascops choliba*) was rescued by the environmental agency and admitted to the wild animals sector of the Veterinary Hospital of the Federal University of Pará.

At the first clinical examination, the animal age was presumed as 11–13 days of life according to the presence of dark plumage and stalks of the remiges appearing, as well as body weight (108 g) [9]. The eyes, nares, oral cavity, choana, and ears were clean, with no exudate, masses, or swellings. The condition of the feathers, body, and feet integument was normal. No parasites were found in the skin. The animal had no beak, wing, or leg fracture or other type of injury. Body condition was determined by palpating the pectoral muscles and scored as 3 (from 1 to 5). Cardiac and pulmonary auscultations were normal. Wing and leg extension, and grip strength, were symmetrical. Bird droppings had white urates and normal feces. The fecal examination was negative for parasites.

Due to the difficulty of finding a conservationist facility for destination and completing hand-rearing, the animal was kept for five months. The animal was fed beef and liver of bovine origin, chicken heart, mice, and *Zophobas morio* larvae, offered twice a day in the proportion of 10% of the owl’s live weight. The meat-based food was supplemented with vitamin A, vitamin B12, vitamin D3, vitamin E, selenium, zinc, copper, phosphorus, and calcium. During this period, the animal was inspected daily to detect behavioral changes; monitor mentation, respiratory distress, and droppings; and confirm food consumption. Routine fecal sample examinations were performed monthly.

After five months of hospitalization and with the clinical condition determined as healthy, the bird began the rehabilitation process for release through falconry techniques to restore the ability to fly and hunt. For this purpose, the bird was equipped with anklets in the tarsus region and other accessories such as straps, swivels, and leashes. The initial phase of falconry training, known as taming, consisted of acclimatizing the animal to handling, the glove, and the habit of feeding off the fist. In the second phase, there was the first jump, where the animal was stimulated by offering food, first on a training perch and then from the trainer’s fist. The third phase, in turn, consisted of a flight stimulus from a fixed point (perch) to the trainer.

The bird started to be fed only during training with mice, chicken, and mealworm larvae, receiving food when landing on the trainer’s glove and repeating the process to improve its physical conditioning. The amount of food provided ranged from 5 to 15 g. The bird was weighed before and after training to check the metabolism of ingested food and body mass. This was associated with the response during physical activity and classified as impaired, fair, good, or excellent.

In the case of positive response to the stimuli, the animal would move on to the following steps: free flight, escape, hunting, and release. The training lasted for 37 days, taking place uninterruptedly, with the bird showing excellent development, responding well to commands for flight, and always appearing attentive and interested in food. It even flew without a guarantor in an external environment with excellent response to the order, with the best flight weight of 97 g.

The training was suspended on the 38th day when the bird showed signs of weakness and lack of appetite. At the beginning of the observation of symptoms, the animal presented progressive weight loss and dyspnea characterized by open-mouth breathing and yawning. Other symptoms were apathy, ruffled feathers, diarrhea, and vomiting. A clinical examination of the animal was performed, which showed a body score of 1, according to Matter et al. [10], with marked loss of pectoral muscles. The nares were normal and symmetric, with no dirt or swelling around the eyes. In the oral cavity examination, secretions and lesions were absent (Figure 1). Tracheal transillumination was performed to search for foreign bodies or parasites; however, solid and liquid materials were absent. The air sacs were auscultated by placing a stethoscope along the lateral and dorsal body walls. No harsh sounds were detected in the air sacs by pulmonary auscultation. Dirty feathers around the cloaca were present. At that moment, a negative result was obtained in the parasitological examination of feces. The methods used were simple sedimentation and flotation in a hyper-saturated sodium chloride solution [11].

The animal was placed into an avian treatment unit cage (ATU, Premium Ecológica) with an oxygen therapy apparatus under heat (37 °C) and humidity (≈90%) control. The oxygen therapy started with about 50% oxygen. Initial treatment was with oxytetracycline (48 mg/kg) every 48 h, intramuscularly; sulfamethoxazole (48 mg/kg), single dose, orally; mebendazole (25 mg/kg), every 12 h for five days, orally; Potenay (0.5 mL/kg), every 24 h for two days, intramuscularly; and 0.9% sodium chloride (50 mL/kg), subcutaneous, together with warmed Mercepton (5 mL/kg) every 8 h for two days for hydration. All procedures were performed under oxygen therapy by mask during the seventh day of treatment, and the owl remained in the ATU at all times. Nutritional support was provided based on the basal metabolic rate (BMR) calculation and maintenance energy requirement (REM) adjusted to 1.5 times TMB. The feeding was forced with the use of ingredients already used routinely in the patient’s diet, supplemented with vitamins as previously described. Fecal collection and copro-parasitological examination were performed using the direct fresh method, according to Hoffmann et al. [11]. However, during subcutaneous fluid therapy, the animal died.

A necropsy was performed according to the techniques of Majó and Dolz [12]. Swabs from the oral cavity and trachea were collected before sample fragment organ collection (liver, proventriculus, small intestine, trachea, gizzard, heart, and skeletal muscles). Samples from the pharyngeal tonsils, tongue, trachea, thoracic and abdominal air sacs, and lungs were aseptically collected for microbiological examination. All samples were stored in sterile falcon tubes and sent in Stuart medium to the Laboratory of the National Primates Center (CENP) for bacterial identification and characterization using VITEK 2 Compact Systems (bioMérieux^®^, Marcy-I’Étoile, France) automated equipment.

The samples used for histopathology were collected from the tongue, pharyngeal tonsils, trachea, thoracic and abdominal air sacs, and lungs contralateral to the previous samples obtained for microbiological examination. Tissue samples were preserved in 10% buffered formalin and sent to the Laboratory of Animal Pathology at UFPA for histological analysis using the hematoxylin–eosin staining technique, as described by Nunes and Cinsa [13]. During the necropsy, fecal samples were collected for parasitological tests using the centrifuge–flotation process described by Faust et al. [14].

This research was authorized by the animal experimentation ethics committee (CEUA) of the Federal University of Pará (UFPA) under protocol number 8888280618 (ID 002193).

## 3. Results

During the observation in the ATU, it was verified that the animal presented apathy, cachexia, ruffled feathers, dirty feathers around the cloaca with dry droppings, vomiting, and dyspnea, spending most of the time with its eyes closed. During this period, a sudden change in the ambient temperature was also observed, as it was a period of intense rainfall with average precipitation of 366 mm and a minimum average temperature of 23 °C with a maximum of 28 °C. The importance of cold environments for birds is emphasized, as they tend to increase their metabolism to maintain body temperature. The copro-parasitological exams carried out using the direct fresh method and the centrifuge–flotation technique detected many eggs of the Diplotriaenoidea superfamily (Figure 1).

Macroscopic findings included severe weight loss with marked loss of pectoral muscle and prominence of the keel, indicating cachexia (Figure 2a). In the oral cavity and proventriculus, pieces of undigested food were found. The cranial and caudal clavicular, thoracic, and abdominal air sacs were thickened, with disseminated turbidity giving a whitish appearance (Figure 2b), and mucopurulent content was detected. Adult nematodes were not observed in the air sacs. The stool was tough in the large intestine with much gas. There was also hepatomegaly, splenomegaly with white splenic discoloration, and marked renal hyperemia.

Bacterial culture showed *Escherichia coli* of bio-number 0405610450406610 with 99% probability, *Enterobacter aerogenes* of bio-number 2607734553576412 with 93% probability, *Klebsiella pneumoniae* of bio-number 6607734453164410 with 98% probability in the tongue, pharyngeal tonsils, trachea, and thoracic and abdominal air sacs. The histopathological examination showed thickening of the air sac wall by amorphous eosinophilic material containing fibrin filaments. The air sac epithelium was reactive, and some inflammatory cells were observed diffusely and lightly distributed throughout the tissue. There was necrotic cellular debris in the aerosacular lumen. Intralesional bacterial colonies were not observed (Figure 3a). At a higher magnification of the air sac, the inflammatory infiltrate was better observed, and was composed of lymphocytes, heterophils, and macrophages, as well as necrotic cellular debris (Figure 3b). The lung showed diffuse hyperemia with granular eosinophilic material deposition in the parabronchi (presence of fibrin) (Figure 3c). Therefore, this bird had moderate multifocal to coalescing heterophilic airsacculitis, likely of bacterial origin. There was no histological evidence of pulmonary nematodes or eggs. No significant histopathological changes were observed in the liver, proventriculus, small intestine, trachea, ventriculus, heart, or skeletal muscles. Thus, the histopathological diagnosis was moderate multifocal to coalescing heterophilic airsacculitis, likely of bacterial origin.

## 4. Discussion

The most frequent helminths in prey birds are nematodes in the digestive and respiratory systems [5,15]. Those belonging to the order Spirurida, family Diplotrianidae, are occasionally found in the air sacs of wild birds, with *Serratospiculum* spp., *Serratospiculoides* spp., and *Diplotriaena* spp. as the prominent representatives responsible for causing intense pulmonary alteration [16,17,18,19,20]. They are described more frequently in Falconiformes than in Strigiformes, which suggests that nematodes are less frequent in the air sacs of this group [16,17,18,19,20]. Recently, there were the first reports of serratospiculiasis in Falconiformes in Latin America [21,22], and it is suggested that this is the first report of the identification of eggs of the superfamily Diplotriaenoidea in *Megascops choliba*.

It should be considered that in parasitic infestations in birds, clinical signs are associated with stressful conditions. Common symptoms caused by parasites of the Diplotriaenoidea family include dyspnea, weight loss, anorexia, and lethargy, in addition to changes in flight performance, thickening of the air sac membrane, and sudden onset of respiratory discomfort [16,17,18,23]. Another important aspect is that adult parasites, larvae, and eggs in the air sacs can damage the tissues and predispose the host to secondary bacterial infections, leading to an increased risk of airsacculitis and pneumonia and resulting in the death of the host [20,24]. In this context, it is feasible to assume that the captive conditions that this owl was subjected to could have contributed a certain degree of stress, leading to immune suppression and subsequently favoring secondary bacterial infections and/or a high parasite load.

Feeding habits directly influence the parasitic fauna, with omnivorous–insectivorous birds being more susceptible to parasitism due to diet diversity [7]. Thus, it should be considered that *Megascops choliba* already has reports of this feeding habit. The parasites of the Diplotriaenoidea superfamily use arthropods, mainly coprophagous beetles, as intermediate hosts in their heteroxenous life cycle. Raptors are usually infected by ingesting intermediate and paratenic hosts [7,24]. In this case, although the bird received a diet based on mealworms, it cannot be said that this was the source of the infection. In addition, hunting birds, even in captivity and within physical barriers, tend to prey on insects that eventually come within their reach.

It is essential to point out that the eggs of the Diplotraenoidea superfamily were identified in the present report only in feces through parasitological examination. Some authors argue that detecting embryonated eggs in the feces does not necessarily indicate the presence of adult nematodes of this superfamily in the air sacs and that a positive copro-parasitological examination would lead to a diagnosis [20,22,23,24]. Considering that the diagnosis most often occurs due to the accidental finding of embryonated eggs during parasitological examinations of feces and pharyngeal swabs, some authors propose the hypothesis of intermittent elimination and recommend the collection and analysis of repeated samples of stool and pharyngeal content as an adequate diagnostic tool [20,24]. It should be added that standard necropsy procedures must be strictly followed to help identify and diagnose these cases [20].

The microbiological examination identified the presence of bacteria belonging to the Enterobacteriaceae family. *Escherichia coli*, for example, is considered commensal and opportunistic; *Klebsiella* spp. and *Enterobacter* spp. are regarded as opportunistic pathogens [25,26]. And, despite being considered commensals in some bird species’ intestinal microbiota, they can multiply and cause intestinal and extra-intestinal infections under favorable conditions. Additionally, histopathological alterations are similar to those observed in enterobacterial infections [26]. Therefore, after correlating this information with the result of 99% probability, it is believed that the primary bacterial agent involved in this case was *E. coli*, since infection by this agent leads to cachexia, lethargy, sepsis, dyspnea, and airsacculitis and is common in immunocompromised animals subjected to stress or overexposure to the agent [26,27].

For antimicrobial therapy, oxytetracycline and sulfamethoxazole were used. Given the impossibility of isolating the agent and performing an antimicrobial sensitivity test, the treatment choice was based on the clinical diagnosis and the rapid evolution of the condition. In this context, oxytetracycline was used due to its good action against Gram-positive and Gram-negative bacteria, and sulfamethoxazole was used due to the initial suspicion of coccidiosis. In the treatment of colibacillosis in *Ara macao*, oxytetracycline hydrochloride (Avitrin antibiotic) was used in a prescription of 5 drops orally every 12 h for seven days, with clinical improvement in the animal at the end of the treatment [28]. However, despite Avitrin being a broad-spectrum antibiotic, the literature points to studies on the antimicrobial resistance of *E. coli*, *Klebsiella* spp., and *Enterobacter* spp. to this drug in wild birds [28,29,30,31], which may have been a factor in the unfavorable clinical evolution of the owl.

Regarding this information, it is essential to emphasize the importance of access to complementary exams while caring for wild animals. Often there need to be sources of funding, making it challenging to send samples. In this case, microbiological tests for isolation and an antibiogram would have particularly contributed to the identification of the agent and allowed for targeted therapeutic implementation, although one should consider the speed of evolution of the pathological condition and the imminent need for intervention. However, the antiparasitic therapeutic scheme included, in addition to sulfamethoxazole, the use of mebendazole, which is reported in the literature as effective in treating parasites of the Diplotriaenoidea superfamily at a dose of 20 mg/kg, administered orally every 24 h for 14 days [32]. Although other authors point out the ineffectiveness of this medication due to the anatomical location of the parasite [18,23], it was likely the best choice among the drugs available at the time of care.

Different treatment protocols have been described with ivermectin (1 mg/kg, intramuscular, single dose), fenbendazole (20 mg/kg, orally, every 24 h for 14 days), doramectin (1 mg/kg, intramuscular, single dose), and merlasomine (0.25 mg/kg, intramuscular for two days) separately or in combination [18,23,32]. However, the treatment recommendation is controversial, as some authors report that the mass of dead parasites in the air sacs can cause necrotic foci. In contrast, others recommend a dose of ivermectin to cause paralysis with later removal of the parasites by endoscopy and a repetition of the drug dose after the procedure [32]. Other studies demonstrate improved flight and fitness after treatment with associated ivermectin and melarsomine [18,23].

## 5. Conclusions

The present report describes the first identification of eggs of the Diplotriaenoidea superfamily in feces and airsacculitis caused by enterobacteria in *Megascops choliba*, which indicates the inclusion of these pathologies as differential diagnoses in respiratory and enteric clinical pictures of *Megascops choliba* and other species of Strigiformes. Furthermore, birds/owls in captivity can deteriorate quickly and lose significant body condition, and, therefore, close monitoring (such as the monitoring this bird received due to flight training) is essential.

## Figures and Tables

**Figure 1 animals-13-02750-f001:**
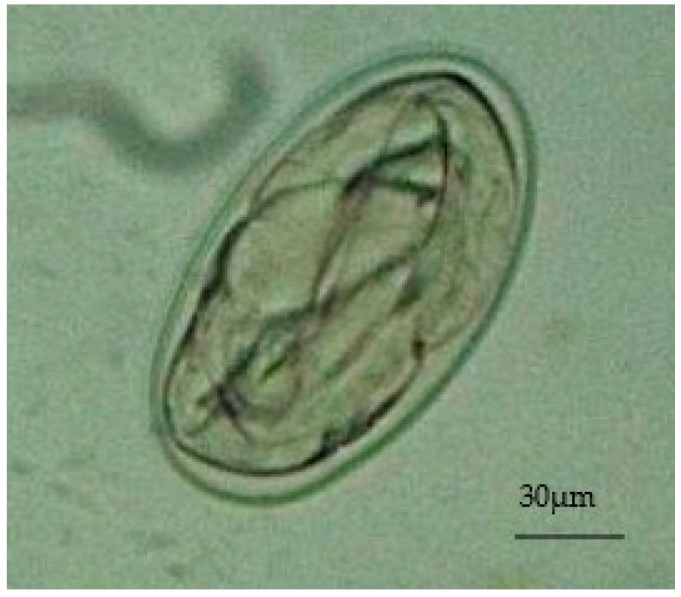
Egg of the Diplotriaenoidea superfamily identified by the direct fresh method, Obj. 40×.

**Figure 2 animals-13-02750-f002:**
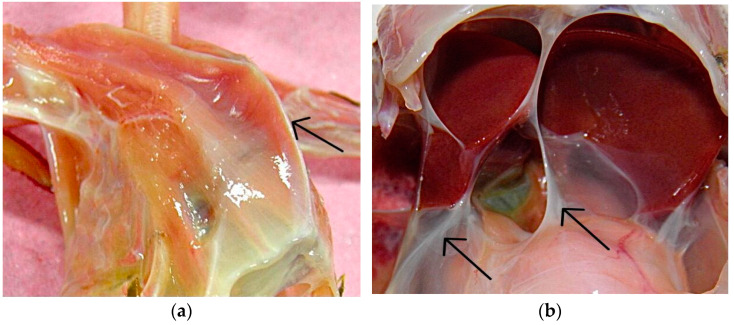
Macroscopic findings: (**a**) Prominence of the keel (black arrow), characterizing severe weight loss; (**b**) thickened caudal thoracic and abdominal air sacs (black arrows) with widespread turbidity and a whitish appearance.

**Figure 3 animals-13-02750-f003:**
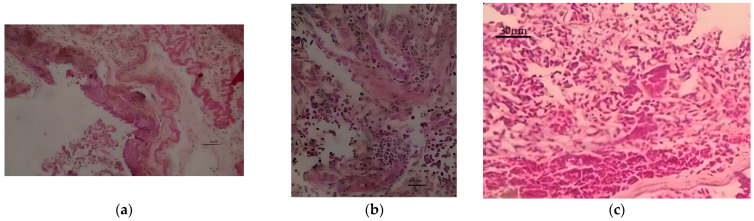
Histopathological examination: (**a**,**b**) Air sac with an area of coalescent multifocal erosion with moderate heterophilic infiltration, Obj. 40×.; (**c**) lung with diffuse and marked hyperemia of the lung tissue. Intravascular thrombus, with the presence of fibrin in air capillaries and marked diffuse hyperemia in blood capillaries, Obj. 40×.

## Data Availability

Not applicable.

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
