# Peer review of "Airsacculitis Caused by Enterobacteria and Occurrence of Eggs of the Superfamily Diplotriaenoidea in Feces of Tropical Screech Owl (Megascops choliba) in the Amazon Biome"

_animals, 2023, doi:10.3390/ani13172750_

Round 1

Reviewer 1 Report (Previous Reviewer 3)

Dear authors, 

Thank you for addressing the points raised on my initial review. I can see that you have made multiple changes to your manuscript, which has provided clarity and significantly improved its scientific strength. I still have one major concern that must be addressed, which is regarding the histopathology. I understand that “the pathology service was provided by a private laboratory”, and that you report exactly what the histopathology report said. However, given that this is a case report that you wish to publish, I recommend that you contact the veterinary pathologist who did the histopathology and ask for clarification on the terminology used on the report, as well as clarification regarding the photomicrographs. Alternatively, you can send your samples to a different pathologist (ideally, with experience on avian pathology), for a second opinion. 

            Regarding the photomicrographs, not only they are of extremely poor quality, but they do not match what the figure legends say. For instance: 

a) Fig 3a: the authors claim that this is an air sac, but it is lung. Also, I can see that you have submitted a total of three images labelled Fig 3A (“Figura 3 A A”, “Figura 3 A” and “Figure 3.A”). All 3 images are from lung, not air sac – I can clearly see air capillaries, which is only present in the lung. Therefore, considering that airsacculitis is one of the main findings of this case report, a proper photomicrograph showing airsacullitis must be added. Moreover, the figure 3a is way too eosinophilic, and the white balance is wrong.

b) Fig 3b: the authors claim that this is a lung with “marked diffuse congestion” with fibrin the “parabronchi”. This is indeed lung. However, the image they added to the paper does not show a parabronchi, but a secondary bronchus. Moreover, there is no evidence of congestion on this image whatsoever. There is a little bit of blood in the airway, but this is likely due to postmortem examination contamination (e.g. when removing the heart) – and does not indicate true pulmonary haemorrhage. I am unable to comment on whether there is fibrin in this bronchus based solely on the images provided. 

Minor points:

-        Title: please add the common name of the owl to the title, alongside the scientific name

-        Abstract: no need to add medication dose/dosage on this section.

-        Line 26: Was the animal ‘bred’ for 5 months? I am not sure if this is a typo or lost in translation, but this does not make sense.

-        Line 29: replace ‘feces’ with ‘droppings’

-        Line 129: replace ‘dirty’ with ‘dirt’ 

-        Line 135: please specify the method of coproparasitological examination used. 

-        Line 141: please specify here the route of fluid therapy, subcutaneous/intraosseous?

-        Line 151: replace ‘necroscopic examination’ with ‘post mortem examination’ or ‘necropsy’

-        Line 152: replace ‘done’ with ‘collected’. I do not understand what you mean by “sample fragment organs collections”.

-        Line 153: replace ‘fragments of’ with ‘samples from’

-        Line 158: On line 199, you mention that liver, proventriculus, ventriculus, small intestine, heart and skeletal muscle did not show histological changes. Please double check this information and correct this here, as needed. 

-        Line 170: replace ‘feces’ with ‘droppings’

-        Line 174: you emphasize here the ‘importance of cold environments’ on avian metabolism. But this information is lost here. If you truly believe that the temperature was too cold for this owl, and that it could have contributed significantly to its decline, then you need to state what is the ideal temperature for this avian species (or, at least, the average temperature for this region at that time of the year) and address this again in your discussion. Alternatively, this comment should be removed.

-        Line 180: replace ‘The impressions of the necroscopic exam were..’ with ‘Macroscopic findings included severe weight loss, with marked loss of pectoral muscle and prominent keel, indicating cachexia’. Add the sex of the bird.

-        Line 186: remove ‘macroscopically’.

-        Line 186: replace ‘The liver…’ with ‘There was also hepatomegaly, splenomegaly with white splenic discolouration, and marked renal hyperemia’.

-        Line 189: Figure legend, replace ‘necroscopic examination’ with ‘post mortem examination’ or ‘macroscopic findings’.

-        Line 192: Replace ‘The microbiology’ with ‘bacterial culture’. Italicise scientific names as appropriate.

-        Line 196: remove ‘also’

-        Line 196-197: this must be corrected. Considering that I did not see the histo slides (or any photomicrograph of the air sac for that matter), I would at least rephrase this sentence to: ‘Histopathological examination of the air sac showed multifocal to coalescing areas of loss of air sac epithelium, associated with moderate heterophilic infiltration.’ Please also add something like ‘intralesional bacterial colonies were not observed’.

-        Line 197: this must also be corrected. As I said earlier, on the photomicrographs you sent, there is no evidence of pulmonary congestion. However, assuming that there was, then this sentence much be at least rephrased to ‘There was diffuse pulmonary congestion with fibrin deposition in the parabronchi’. You must include a sentence on whether or not there was any histological evidence of pulmonary nematodes or eggs.

-        Line 199: On ‘Liver….’, rephrase to ‘No significant histopathological changes were observed on the liver, …”. Also, please replace ‘gizzard’ with ‘ventriculus’. 

-        Line 201: a histopathology report will never say ‘bacterial airsacculitis’ unless there was bacterial colonies in the air sac. Therefore, rephrase this to ‘Therefore, this bird had moderate multifocal to coalescing heterophilic airsacculitis, likely of bacterial origin‘. Also, please standardize the terminology ‘airsacculitis’ vs ‘aerosacculitis’

-        Line 217: this initial sentence does not make sense. Please rephrase. Replace ‘stress’ with ‘stressful’.

-        Line 224: this last sentence is very important, but it is difficult to understand and it must be rephrased. Please use ‘this owl’ instead of ‘the specimen in this study’. Suggestion: ‘In this context, it is feasible to assume that the captive conditions that this owl was subjected to could have contributed to a certain degree of stress, leading to immune suppression, and subsequently favouring secondary bacterial infections and/or higher parasite load’.

-        Line 240: ‘stool test’ is extremely informal, and unsuitable for a scientific manuscript. Rephrase it to ‘coproparasitological examination’. Also, ‘close the diagnosis’ is not a terminology used in English. Perhaps you mean ‘reach a diagnosis’? 

-        Line 144: ‘This is added’ does not make sense. Please rephrase. 

-        Line 280: ‘Still on the possibilities’ does not make sense. Please rephrase. 

-        Conclusions: This is literally a 1-sentence conclusion. Could you please expand this a little more, perhaps separating this into at least 2 sentences? Perhaps you can highlight that birds/owls in captivity can deteriorate quick and lose significant body condition, and, therefore, close monitoring (such as the one this bird was having due to flight training) is essential. 

-        Line 300: There is a double semi colon (;;)

-      Lines 312, 313, 314, 316: replace ‘by’ with ‘for the’

The authors believe that the English language used throughout the manuscript is of adequate quality for a scientific manuscript. However, I respectfully disagree. Under ‘minor points’ I highlight multiple occasions where the sentence they wrote does not make sense in English. Therefore, I still strongly recommend that the authors seek the help of a native speaker and/or a professional English language editing service. 

Author Response

Dear reviewer 1, thank you very much for once again dedicating your time to improving this manuscript. Thank you very much.
Sincerely,

Felipe Masiero Salvarani

Reviewer 2 Report (Previous Reviewer 2)

The authors have addressed my comments, thus I recommend this paper to be accepted in the present form.

Sincerely

V

For future submissions, it is recommended to contract english proofreading services to accelerate manuscript revision, and acceptance. 

Author Response

Reviewer 2

We would like to thank reviewer 2 for reviewing our manuscript again and stating that "The authors have addressed my comments, thus I recommend this paper to be accepted in the present form". Thank you very much.

Sincerely,

Felipe Masiero Salvarani

Reviewer 3 Report (Previous Reviewer 1)

Dear Authors, As reported in my previous review report, I confirm the outcome of my evaluation as "Accept in present form", renewing my best wishes for wide dissemination and approval of the paper by the international scientific community.

Sincerely

Author Response

Reviewer 3

We would like to thank reviewer 3 for reviewing our manuscript again and stating once again that "Dear Authors, As reported in my previous review report, I confirm the outcome of my evaluation as "Accept in present form", renewing my best wishes for wide dissemination and approval of the paper by the international scientific community." Thank you very much.

Sincerely,

Felipe Masiero Salvarani 

Round 2

Reviewer 1 Report (Previous Reviewer 3)

Dear authors, 

Thank you for taking on board my comments and suggestions. You have greatly improved the manuscript, especially the pathology section. My only concern is that the histo images are still of extremely poor quality. If the pathology lab cannot locate the paraffin block for a new recut, then new photomicrographs using the original histology slides should be taken. Please seek the help of a different institution, if necessary, if a microscope with camera (or digital scanner) is not available in your institution or at the pathology lab. Perhaps the veterinary pathologist and now co-author can advise you on that. And the white balance must be corrected before these images are of enough quality for a scientific publication.

Minor points:

-       Title: use the English common name of the owl: tropical screech owl. And leave the scientific name in brackets. 

-       Fig 3A: Replace ‘neurotic debris’ with ‘necrotic cellular debris’. The word ‘neurotic’ is incorrectly used a few times across the text, at least. 

-       Fig 3B: English must be corrected. ‘Higher’ not highest. Remove ‘mainly’. Use ‘heterophils’, not ‘granulocytes’. Use ‘as well as’ instead of ‘in addition to being possible to observe the presence of’. Replace ‘neurotic cells’ with ‘necrotic cellular debris’

-       Fig 3C. This picture is absolutely out of focus. This image must be removed, if it can’t be replaced. 

-       Fig 3D. Remove ‘the’ before ‘air capillaries’ and before ‘blood capillaries’

Apart from the typos (e.g. neurotic vs necrotic), the English quality is reasonable. The only problem is that some sentences are a literal translation of the Portuguese, with incorrect prepositions being used, which makes the sentences difficult to read.

Author Response

Reviewer 1

We would like to thank reviewer 1 for reviewing our manuscript again and recognizing our efforts to follow his suggestions and make the manuscript better.

With regard to histopathology, we accept your criticisms. However the laboratory that carried out the histopathological service did not have a backup that could be used to create new histological sections and new photomicrographs. Also, due to our inexperience, we do not keep a backup of the samples sent for histopathology. Unfortunately, in addition to not having a backup of paraffin block for a new recut, we are also on an academic vacation here in Brazil, which makes new photomicrographs unfeasible, but we removed the worst of the photos, as per your suggestion which was the 3C (the 3D figure became figure 3C), as can be seen in the final version of the manuscript.

Minor points:

  • Title: use the English common name of the owl: tropical screech owl. And leave the scientific name in brackets. Changed as per your request.
  • Fig 3A: Replace ‘neurotic debris’ with ‘necrotic cellular debris’. The word 'neurotic' is incorrectly used a few times across the text, at least. Changed as per your request.
  • Fig 3B: English must be corrected. ‘Higher’ not highest. Remove 'mainly'. Use 'heterophils', not 'granulocytes'. Use 'as well as' instead of 'in addition to being possible to observe the presence of'. Replace ‘neurotic cells’ with ‘necrotic cellular debris’. Changed as per your request.
  • Fig 3C. This picture is absolutely out of focus. This image must be removed, if it can't be replaced. Removed as directed.
  • Fig 3D. Remove 'the' before 'air capillaries' and before 'blood capillaries'. Changed as per your request.
  • Comments on the Quality of English Language: Thank you for your rating as "Minor editing of English language required". In the final review of the manuscript, the journal itself and its editors correct some phrases, sentences and expressions that will certainly make the article perfect for reading in English.

Sincerely

Felipe Masiero Salvarani

This manuscript is a resubmission of an earlier submission. The following is a list of the peer review reports and author responses from that submission.

Round 1

Reviewer 1 Report

Dear Authors,

I congratulate you for the topic covered and for the novelty it brings in the context of the clinic of non-conventional animals and veterinary epidemiology. I am sure your study will have a major impact on future research in this rapidly growing area of research.

The description of the case report is well structured and full of images that make reading easy and demonstrative of the work done. Therefore, I feel that I can approve your manuscript in its present form.

Kind Regards

Author Response

Reviewer 1

We would like to thank Reviewer 1 for his remarks. The recognition of the originality of this work and its contribution to the scientific community is a great honor. As we read your words "Dear Authors, I congratulate you for the topic covered and for the novelty it brings in the context of the clinic of non-conventional animals and veterinary epidemiology. I am sure your study will have a major impact on future research in this rapidly growing area of research. The description of the case report is well structured and full of images that make reading easy and demonstrative of the work done. Therefore, I feel that I can approve your manuscript in its present form.", we have a sure that all the effort and dedication involved in this manuscript was rewarding and we feel honored.

Best regards,

Felipe Masiero Salvarani

Reviewer 2 Report

The authors did a great job describing this novel case. There are some minor editing in the paper, and few comments I have. 

Line 242 - Should say foci instead of focus. 

Describe what kind of histopathological staining technique was performed.  

Good quality of english. 

Author Response

Reviewer 2

We would like to thank Reviewer 2 for his remarks. We inform you that we have made the modification to line 242 foci and not focus and also, we describe what kind of histopathological staining technique was performed (using the hematoxylin-eosin staining technique). And we are honored to read your comment on the quality of our work by saying "The authors did a great job describing this novel case". Thank you very much.

Best regards,

Felipe Masiero Salvarani

Reviewer 3 Report

The authors present a case report of bacterial airsacculitis and faecal Diplotrianeoidea eggs in a Tropical screech owl. Both conditions, bacterial airsacculitis and Deiplotrineoidea infestation, have been extensively reported in birds. The novelty of this manuscript is that it represents the 1st report of Deiplotrineoidea infestation in this owl species. 

In my opinion, the three major issues of this manuscript are: a) the title is misleading, b) the conclusions are inconsistent with the data presented, and c) the clinical management during hospitalization was very poor. The manuscript is poorly written and there is key information missing from the manuscript, making it scientifically weak.

1)     Misleading title: Based on the title, it sounds that this bird had airsacculitis caused by enterobacteriacea and by eggs of Diplotrianeoidea - however, this is incorrect. As the authors explain on the main text, Enterobacteriacea were isolated from the “tonsils” and tongue of this owl, and, and Diplotrianeoidea eggs were only detected on faecal examination (not on air sacs). The only way the authors can claim that the airsacculitis was caused by enterobacteriacea, was if they had isolated enterobacteriacea directly from the air sacs - but they did not sample the air sac for microbiology. Similarly, they did not find any adult nematode within the respiratory tract on pot-mortem examination, and they did not find any histological evidence of Diplotrianeoidea infestation within the respiratory tract (trachea/lungs/air sacs).

2)     Unclear samples used for microbiology: The authors state that “tongue and tonsils were collected” for microbiology, and that swabs “from the oral cavity” were used for microbiology. This needs clarification. The authors must clarify whether they refer to pharyngeal or caecal tonsils. Did they swab the tongue and pharyngeal tonsils after they were collected? Or they removed the tongue and pharyngeal tonsils and then swabbed the oral cavity? 

3)     Inconsistent microbiology results and interpretation: Assuming that the sample submitted for microbiology was an oral swab and/or tongue/pharyngeal tonsil (it is not clear on the text), how can the authors assume that the bacteria isolated was a primary pathogen and not a simply commensal or a sampling contaminant? Can the authors explain why air sac swab was not collected for bacteriology? If bacterial colonies were not observed associated with the heterophilic airsacculitis, how can the authors conclude that the heterophilic airsacculitis was caused by bacteria? Is there any evidence that this bird was in sepsis?

4)     Very poor clinical management during hospitalization: the authors claim that when this bird was admitted to the veterinary hospital, it was unable to fly or hunt due to its young age, but “no fracture of other type of injury was found”. However, they do not add any clinical examination finding (e.g., including mental state, body condition, hydration, plumage condition, etc.) or whether or not they performed any ancillary test (e.g. blood work/biochemistry, x-ray, faecal examination, etc). This indicates a very poor animal health care upon initial presentation. Similarly, the authors state that this bird started showing clinical signs on day 38thof training – despite the clinical signs, no physical examination was performed. The bird was then kept for 2 days “under observation” until empirical treatment started – again, no mention of any physical examination. And on the 7th day after the bird started showing clinical signs, a direct faecal exam was performed. And again, no mention whatsoever of any physical examination being performed. During post-mortem examination, it was noted that this bird was in extremely poor body condition – if a physical examination was performed on this patient while it was still alive, the poor body condition would be noted, and this would be addressed (e.g., the use of a critical care/recovery formula), and ancillary test would have been performed. This highlights the very poor clinical management of this patient, from its admission do the point of death, raising serious ethical concerns. 

5)     Poor and incomplete macroscopic/histopathological description: The paragraph of post-mortem examination findings is very brief and does not include what organs were normal. In this case, it is extremely important to include whether any lesions were observed in the respiratory tract (sinuses, trachea, bronchi, lungs, and air sacs). They must also specify what air sacs were inspected on post-mortem examination. 

Regarding histopathology, the authors very briefly mention 2 histological changes, heterophilic infiltration in air sacs and fibrin in the parabronchi; nevertheless, they only give 1 morphological diagnosis (heterophilic airsacculitis). The authors must give clear explanation on why the trachea and bronchi were not sampled for histopathology. They must also include a list of every tissue examined histologically, even if no significant histological changes were observed. As a minimum, the authors must perform a Gram stain on the relevant tissues. 

Although this is not a pathology journal, the terminology used to describe macroscopic (e.g., hepatomegaly, splenomegaly) and microscopic findings (e.g., fibrin is fibrillary, not granular; air sac is lined by a single layer of epithelium and, therefore, do not erode, but ulcerate), must be reviewed. The authors seem to use the terms “hyperaemic” and “congested” interchangeably.

6)     Figures: Macroscopic images are relevant and of good quality. However, photomicrographs need to be improved. Figure 3a: too eosinophilic; cannot recognise tissue as being air sac. Figure 3b: it does not seem a parabronchi (perhaps a secondary bronchi?); there is no evidence of congestion in this image (only some extravasated blood in lumen).

This manuscript requires extensive editing of English language. I suggest the use of a professional English language editing service. 

Author Response

Reviewer 3

We would like to thank Reviewer 3 for his remarks. Despite the fact that at first I read how emphatic he was when he stated that in his opinion "three major issues of this manuscript are: a) the title is misleading, b) the conclusions are inconsistent with the data presented, and c) the clinical management during hospitalization it was very poor. The manuscript is poorly written and there is key information missing from the manuscript, making it scientifically weak." We use your questions to scientifically improve the work which I report was highly praised for its originality, scientific content and writing in English (by Latinos), by reviewers 1 and 2. As there were many "suggestions" by reviewer 3, we preferred to make a table to show the changes and also express our opinion. Anyway thank you very much for your contributions, I'm sure the article was much better after your criticisms.

Reviewer 3 comments to author

Authors answer

1) Misleading title: Based on the title, it sounds that this bird had airsacculitis caused by enterobacteriacea and by eggs of Diplotrianeoidea - however, this is incorrect.

We agree with the suggestion and changed the title as follows:

Airsacculitis caused by enterobacteria and occurrence of superfamily Diplotriaenoidea eggs in the feces of a Megascops choliba from Amazon Biome.

All text was corrected following the review's suggestion.

We really had mistakes in the translation of the text, and important details were lost.

As the authors explain on the main text, Enterobacteriacea were isolated from the “tonsils” and tongue of this owl, and, and Diplotrianeoidea eggs were only detected on faecal examination (not on air sacs). The only way the authors can claim that the airsacculitis was caused by enterobacteriacea, was if they had isolated

enterobacteriacea directly from the air sacs - but they did not sample the air sac for microbiology.

The Enterobacteriacea was detected in the air sacs.

We noted that the information about samples sent for microbiology was incomplete in the primarily text. Note that all samples sent are now related in materials and methods.

Similarly, they did not find

any adult nematode within the respiratory tract on pot-mortem examination, and they did not find any histological evidence of

Diplotrianeoidea infestation within the respiratory tract (trachea/lungs/air sacs)

We agree with the Reviewer suggestion.

Thus, we corrected the text reporting the occurrence of superfamily Diplotriaenoidea eggs in the feces of a Megascops choliba.

Unclear samples used for microbiology: The authors state that “tongue and tonsils were collected” for microbiology, and that swabs “from the oral cavity” were used for microbiology. This needs clarification. The authors must clarify whether they refer to pharyngeal or caecal tonsils. Did they swab the tongue and pharyngeal tonsils after they were collected? Or they removed the tongue and pharyngeal tonsils and then swabbed

the oral cavity?

We agree with the Reviewer, the text was corrected.

Note that in lines 157-162 there is a complete explanation about samples for microbiology.

“Swabs from the oral cavity and trachea were done before sample fragment organ collections. Fragments of pharyngeal tonsils, tongue, trachea, thoracic and abdominal air sacs, and lungs were aseptically collected for the microbiological examination. All samples were stored in sterile falcon tubes and sent in Stuart medium to the Laboratory of the National Primates Center (CENP) for bacterial identification and characterization using VITEK Compact II (bioMérieux®) automated equipment.”

Inconsistent microbiology results and

interpretation: Assuming that the sample submitted for microbiology was an oral swab and/or tongue/pharyngeal tonsil (it is not clear on the text), how can the authors assume that the bacteria isolated was a primary pathogen and not a simply commensal or a sampling contaminant? Can the authors explain why air sac swab was not collected for bacteriology? If bacterial

colonies were not observed associated with the heterophilic airsacculitis, how can the authors conclude that the heterophilic

airsacculitis was caused by bacteria? Is there any evidence that this bird was in sepsis?

 Air sacs were collected for microbiological analysis as reported in materials and methods. We emphasize that this result motivated the production of the paper.  We noted that the information about samples sent for microbiology analysis were incomplete in the primary text and proceed the correction in this English version.

Very poor clinical management during

hospitalization: the authors claim that when this bird was admitted to the veterinary hospital, it was unable to fly or hunt st due to its young age, but “no fracture of other type of injury was found”. However, they do not add any clinical examination finding (e.g., including mental state, body condition, hydration, plumage condition, etc.) or whether or not they performed any

ancillary test (e.g. blood work/biochemistry, x-ray, faecal examination, etc). This indicates a very poor animal health care upon initial presentation. Similarly, the authors state that this bird

started showing clinical signs on day 38 of training – despite the clinical signs, no physical examination was performed. The bird was then kept for 2 days “under observation” until empirical treatment started – again, no mention of any physical

examination. And on the 7 day after the bird started showing clinical signs, a direct faecal exam was performed. And again, no

mention whatsoever of any physical examination being performed. During post-mortem examination, it was noted that this bird was in extremely poor body condition – if a physical examination was performed on this patient while it was still alive, the poor body condition would be noted, and this would be addressed (e.g., the use of a critical care/recovery formula), and ancillary test would have been performed. This highlights the very poor clinical management of this patient, from its admission do the point of death, raising serious ethical concerns.

Unfortunately, during the translation of the text, important details were suppressed, mainly the chronology of events, which compromised the explanation of the patient's clinical management. Details of the patient’s physical examination and management were complemented in the text so that the events are better understood. Please, see materials and methods.

About performing complementary exams it is important to elucidate that at the time of this patient admission, different from nowadays, there were no access to specialized team in the institution for avian hematological exam. Furthermore there was no access to perform an image exam in the institution since there was no radiographic equipment of sufficient accuracy for the size of the animal. Considering that there were no financial support for sending samples to other specialized laboratories, the team of veterinarians did all that was possible in that time to treat the specimen. In this context, the team contacted some other institutions trying to get partnership for complementary exams and received a positive answer only for microbiological analysis in Laboratory of the National Primates Center (CENP). Even so, it was not possible to carry out an antibiogram.   

Poor and incomplete macroscopic/histopathological description: The paragraph of post-mortem examination

findings is very brief and does not include what organs were normal. In this case, it is extremely important to include whether

any lesions were observed in the respiratory tract (sinuses, trachea, bronchi, lungs, and air sacs). They must also specify what air sacs were inspected on post-mortem examination. Regarding histopathology, the authors very briefly mention 2 histological changes, heterophilic infiltration in air sacs and fibrin in the parabronchi; nevertheless, they only give 1 morphological diagnosis (heterophilic airsacculitis). The authors must give clearexplanation on why the trachea and bronchi were not sampled for histopathology. They must also include a list of every tissue examined histologically, even if no significant histological changes were observed. As a minimum, the authors must

perform a Gram stain on the relevant tissues. Although this is not a pathology journal, the terminology used to describe macroscopic (e.g., hepatomegaly, splenomegaly) and microscopic findings (e.g., fibrin is fibrillary, not granular; air sac

is lined by a single layer of epithelium and, therefore, do not erode, but ulcerate), must be reviewed. The authors seem to use

the terms “hyperaemic” and “congested” interchangeably

We keep the macroscopic and histopathological findings as described in the animal prontuary.

Figures: Macroscopic images are relevant and of good quality. However, photomicrographs need to be improved. Figure 3a: too eosinophilic; cannot recognise tissue as being air sac. Figure 3b: it does not seem a parabronchi (perhaps a secondary bronchi?); there is no evidence of congestion in this image (only some extravasated blood in lumen).

The pathology service was carried out by a private laboratory and the data presented by them are what we have to compose the article. I'm sorry if our knowledge of pathology is not at the level you would like, but we present what was possible and what was passed on to us by the laboratory.

This manuscript requires extensive editing of English language. I suggest the use of a professional English language editing service.

The other two reviewers did not share this opinion, nor do we think that the English writing is so bad as to disqualify the manuscript. But we respect your opinion and will leave it up to the editors to decide.

Best regards,

Felipe Masiero Salvarani
